# Contemplation of Improvement Efforts to Manage Combined Sewer Overflows †

**Younghan Edwin Jung** [1],*, **M. Myung Jeong** [2], **Hwandon Jun** [3] **and Trevor Smith** [4]

1. Department of Construction, Seminole State College, Sanford, FL 32746, USA
2. Department of Civil Engineering and Construction, Georgia Southern University, Statesboro, GA 30458, USA; mjeong@georgiasouthern.edu
3. Department of Civil Engineering, Seoul National University of Science and Technology, Seoul 01811, Republic of Korea; hwjun@seoultech.ac.kr
4. NuScale Power, Ocean Springs, MS 39564, USA; trevor.gerald.smith@gmail.com
* Correspondence: jungy@seminolestate.edu
† This paper was originally published at the 55th ASC Annual International Conference in Denver, CO, USA. It has been extended and modified for publishing in the journal.

**Abstract:** Combined sewer overflow (CSO) is a significant environmental concern and public health risk (e.g., water contamination, eutrophication, and beach closure). The Environmental Protection Agency (EPA) has introduced the National Pollutant Discharge Elimination System (NPDES) permitting program to regulate and address this matter. This program mandates the control of CSOs for more than 700 municipalities obligated to devise Long-term Control Plans (LTCPs) to curb combined sewer overflows and reduce them to safe levels. The LTCP involves diverse strategies, including sewer separation, green infrastructure improvements, and conventional gray infrastructure upgrades. This study investigates several municipalities' solutions for CSO problems that use conventional methods and wireless sensor technology as real-time control, mainly focusing on a comparative analysis of two cities, Richmond, Virginia, and South Bend, Indiana, such as their average rainfall, the frequency of overflows, and the capacity of treatment plants. The findings indicate that integrating sensor technology could significantly enhance modeling endeavors, bolster the capacity of existing structures, and substantially enhance preparedness for storm events. The EPA's Storm Water Management Modeling (SWMM) software is utilized. Through an analysis of SWMM data, the study suggests the potential for leveraging wireless sensor technology to achieve more robust control over CSOs and significant cost savings as a part of LTCPs.

**Keywords:** combined sewer overflow (CSO); long-term control plan; wireless sensor technology; stormwater management; stormwater management modeling (SWMM)

## 1. Introduction

A combined sewer system (CSS) is a sewage infrastructure that collects and conveys both domestic sewage and stormwater runoff in a single-pipe system. This practice was common in older towns and urban areas before developing separate sewage systems (SSS). In a CSS, the same pipe carries all residential, commercial, and industrial sewage as well as rainwater from streets, roofs, and other impermeable surfaces during rain. CSSs were designed over 150 years ago to convey wastewater and stormwater directly into waterways. Due to blockages or high stormwater flow, wastewater frequently overflows and is directed to the natural environment with little or no treatment. This is called a combined sewer overflow (CSO). While the operation of CSOs sometimes prevents houses and businesses from flooding, the introduction of polluted water into the waterbody has the potential to pose a hazard to the environment [1].

In 2023, about 700 municipalities in the US have CSOs [2,3]. A total of 9348 CSO outfalls are identified and regulated by National Pollutant Discharge Elimination System

(NPDES) permits [3]. Combined sewer systems are found in 32 states (including the District of Columbia) and nine EPA Regions. CSO communities are regionally concentrated in older communities in the Northeast and Great Lakes regions. Figure 1 shows a typical CSS compared to an SSS. SSSs have two separate pipes. One pipe carries stormwater (rainwater) from storm drains to local streams. Pollution and trash in stormwater flow to local waterbodies with little or no treatment. A second pipe carries sanitary sewage to the wastewater treatment plant (WWTP). When it rains in a CSS, stormwater flows into the same pipe and mixes with raw sewage. In dry weather, all sewage flows to the WWTP. The combined volume of stormwater becomes significant, particularly during periods of heavy rainfall. For instance, in Alexandria, Virginia, USA, there can be nine times more stormwater than raw sewage when it rains [4]. Subsequently, the stormwater can overwhelm the CSS. When this happens, the mixture may overflow into local streams through one of the four permitted combined sewer outfalls in Alexandria. Permitted outfalls are located throughout the system to act as relief points during wet weather. These outfalls discharge untreated or partially treated stormwater and wastewater into nearby waterbodies. The volume of stormwater and wastewater entering the system can exceed its capacity to treat and manage the water. There will likely be a higher frequency and intensity of extreme rainfall in the future, leading to an increased risk of urban flash floods [5]. To prevent backups and flooding, these permitted outfalls provide a controlled release mechanism. The EPA estimates that over 850 billion gallons of untreated wastewater and stormwater are released as CSO each year from over 750 cities in the US [3].

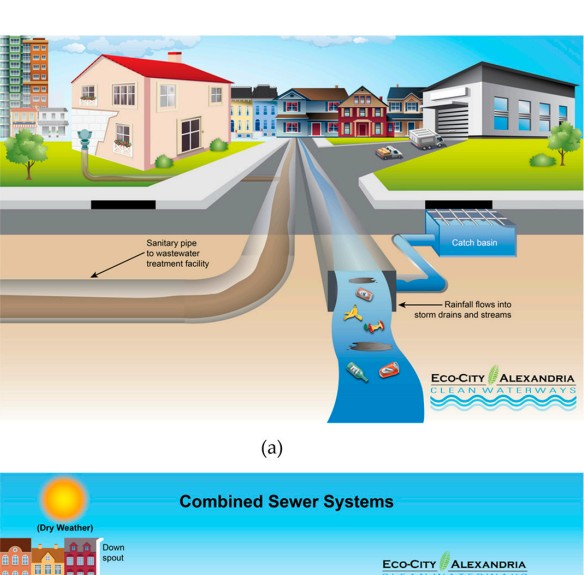

(a)

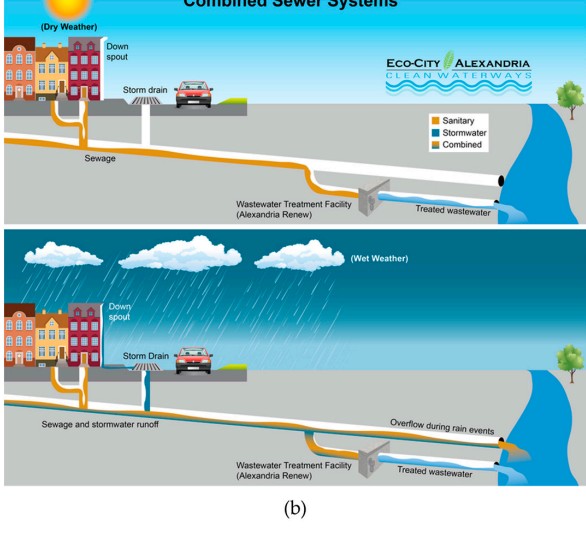

(b)

**Figure 1.** Illustration of CSS and SSS. (**a**) Separate Sewer System (**b**) Combined Sewer System Reprinted/adapted with permission from Ref. [4].

Similar to the US, many countries' existing sewer networks are not designed to handle the collective stormwater and wastewater separately during stormy periods [1]. Because of this capacity limit, overflows frequently occur. CSOs have hazardous consequences for surface waters, both health-related and economic risks, if they are not adequately controlled [6–9]. According to the EPA report to Congress in 2004, CSOs and sanitary sewer overflows (SSOs) cause high levels of E. coli bacteria, leading to harmful algal blooms and between 3500–5500 gastrointestinal illnesses yearly [3]. It can also have large economic impacts when recreational areas have to be shut down or avoided.

Due to the extreme costs of preventing these overflows, many cities did not confront these problems until the late 20th century. In 1994, brought on by violations of the Clean Water Act, the EPA issued the CSO Control Policy. Through the use of the NPDES permitting program, cities were mandated to immediately reduce and plan to eliminate CSOs or face significant fines. As a result, cities had to present Long-term Control Plans (LTCP) to prevent CSOs. Moreover, ongoing climate variability and climate changes may cause intensified precipitation events in some areas, which may also lead to frequent CSOs [9–13]. Stormwater Detention Tanks (SWDTs) are a prevalent conventional approach employed in numerous nations to address urban catchment flooding issues [14]. The Precipitation Variability Adaptation Index (PVAI) emerged as a straightforward instrument to evaluate how Sustainable Urban Drainage Systems (SuDS) can help urban areas adapt to the hydrological impacts of potential climate shifts [15]. In essence, a significant drawback of the conventional system is its inability to effectively segregate contaminated wastewater from the combined water during periods of intense precipitation.

Therefore, minimizing CSOs in urban areas is an essential task for many municipal councils [9]. Many methods to control CSOs as part of their LTCP fulfill a commitment to the EPA to lessen the amount of wastewater that ends up in rivers and streams. The manageable methods accepted by the EPA are separation, green infrastructure, and gray infrastructure. These approaches can be classified into structural and non-structural measures. The structural measures for controlling CSOs, such as underground tunnels to store combined sewer flows on stormy days, are physical constructions. Otherwise, non-structural measures utilize knowledge and experiences to develop various policies and control approaches to reduce the CSOs in existing sewer networks [9]. The financial capabilities and disturbances to the inhabitants have limited the structural measures for minimizing the CSOs [9,13]. Therefore, non-structural measures, such as real-time monitoring, modeling, data processing and analytics, and artificial intelligence/machine learning, which provide dynamic feedback and optimization, are a higher priority today. [9,16]. For instance, new research and declining costs in wireless sensor technology have allowed for an alternative solution using wireless flow and level sensors to monitor and manipulate stormwater flow. It is reported that real-time control plays a significant role in sewer network control [9].

## 2. Objective and Scope of Study

This study investigates the benefits and shortcomings of conventional methods (i.e., separation, green infrastructure, and gray infrastructure) and wireless sensor technology as real-time control by using information gained from EPA reports and local public works sources. Much of this data, including annual overflow and system effectiveness, was obtained by the cities' use of the EPA's Storm Water Management Modeling (SWMM) software. We selected two cities, Richmond, Virginia, and South Bend, Indiana, to investigate CSO problems. The two cities have had similar problems, including average rainfall, overflows, and treatment plant capacity. South Bend is taking a more modern approach, using inexpensive wireless sensor technology to enhance modeling efforts, increase capacity in the existing structures, and better prepare for storm events. In contrast, Richmond is focused on traditional methods, using primarily gray and green infrastructure improvements, along with monitoring and modeling. Integrating intelligent control systems to optimize the current infrastructure's capacity is considered an efficient and uncomplicated approach for mitigating peak runoff and CSOs while maintaining cost savings. After learning from

the investigation of two cities, this study aims to use available SWMM data to suggest the best path to effective management for CSO issues in Richmond, VA.

## 3. Conventional Methods

The primary objective of the city's CSO prevention strategy in the conventional CSS revolves around efficiently redirecting a substantial portion of both stormwater and wastewater to the WWTP during the peak periods of wastewater daily usage and annual stormwater cycles. To understand these complex systems, cities rely on standardized modeling tools such as the SWMM to gain insight into stormwater and wastewater inflows. These sophisticated models allow municipalities to assess handling capacity and predict potential flooding. SWMM leverages geographic and sewage system data to effectively measure and predict the impact of storm events on the entire system. SWMM data suggests that the solution to preventing CSOs is creating or allowing for more capacity within the system or lowering the flow of stormwater that enters the system. SWMM data has also allowed for the research of the following three major categories for controlling CSOs: total separation of the wastewater system, green infrastructure, and gray infrastructure.

The first approach is to increase the system's capacity by creating new infrastructure or allowing for expansion. This approach aims to reduce the potential for flooding during peak periods by accommodating larger volumes of water. A second way to control CSOs is through the implementation of green infrastructure. It entails integrating green elements such as permeable surfaces, rain gardens, and green roofs. These practices promote natural rainwater absorption, minimizing strain on the sewer system, and reducing overflow. The third category entails building gray infrastructure consisting of conventionally engineered solutions such as storage tanks, tunnels, and cisterns. These structures effectively manage stormwater and wastewater, mitigating the risk of combined sewer overflow. SWMM's capabilities allow cities to thoroughly investigate and evaluate these three control strategies. Understanding the potential benefits and limitations of complete segregation, green infrastructure, and gray infrastructure can help cities develop comprehensive CSO prevention plans and move toward more sustainable and resilient urban environments.

### 3.1. Separation

Sewer separation can be accomplished by installing a new stormwater or sanitary sewer alongside the existing sewer. The main factors determining the use after separating the existing lines are the economics, capacity, conditions, and arrangement of the combined sewer pipe. EPA's SWMM is used worldwide for planning, analysis, and design related to stormwater runoff, combined and sanitary sewers, and other drainage systems [17]. Figure 2 shows a graph developed by the EPA [18] using various SWMM data to illustrate how various combinations of increased gray infrastructure storage capacity and green methods impact CSOs. The EPA [17] developed a hypothetical case study to illustrate how a community might use Hydrologic and Hydraulic (H&H) modeling, as shown in Figure 2a. Hydrologic indicates where rainwater goes and how much will flow into the sewer network, while hydraulics indicates the volume and velocity of flow in the sewer network. Based on H&H modeling, one can estimate the CSS's performance and resultant CSO event frequencies and discharge volumes similar to the ones depicted in Figure 2b. Figure 2a also shows a map of a hypothetical CSS that covers a 500-acre service area. From the results of Figure 2, the city might determine that adding 1.6 million gallons of underground storage along with a robust green infrastructure plan is more cost-effective than simply adding 3 million gallons of underground storage with the same results.

The method used for total prevention of CSOs is the complete separation of the sewer system. For example, the Department of Public Works in Grand Rapids, MI, eventually used this method to solve its CSO problem. Grand Rapids is the second-largest city in the state of Michigan and serves as the county seat of Kent County. At the 2020 Census, the city had a population of 198,917, just under 200,000. [19]. The city finished converting its combined sewers to separate ones in 2015. During the 1960s, 59 points in the city were

identified where raw or partially treated sewage could overflow into a nearby waterbody (the Grand River). A total of 12.6 billion gallons of raw, untreated sewage flowed into the Grand River in 1969. The city spent USD 2.7 million on wastewater treatment in 1991 and a USD 1.2 million underground storage facility in 2015. They were able to lower CSOs from 12 billion gallons in the 1970s to zero CSO events in 2014. The separation started in 1991 and cost the city USD 400 million [20]. The most significant hurdle for separation is the financial means to construct such a system with limited financial sources. Over 700 cities in the US affected by CSOs have populations of less than 10,000 people [3]. The costs associated with separation projects are overwhelming and unfeasible for small cities such as Detroit, with larger populations and decreasing budgets. One of the largest sewer separation projects is underway in London. It will prevent untreated sewage from entering the existing Victorian sewer network, designed to serve 4 million people [21]. The Thames Tideway Tunnel is a large-scale sewer to serve 16 million people by 2160. This project will be completed approximately 16 miles (25 km) and 25 feet (7.5 m) in diameter in 2025 [22] and will cost USD 4.5 billion (GBP 3.5 billion) [23].

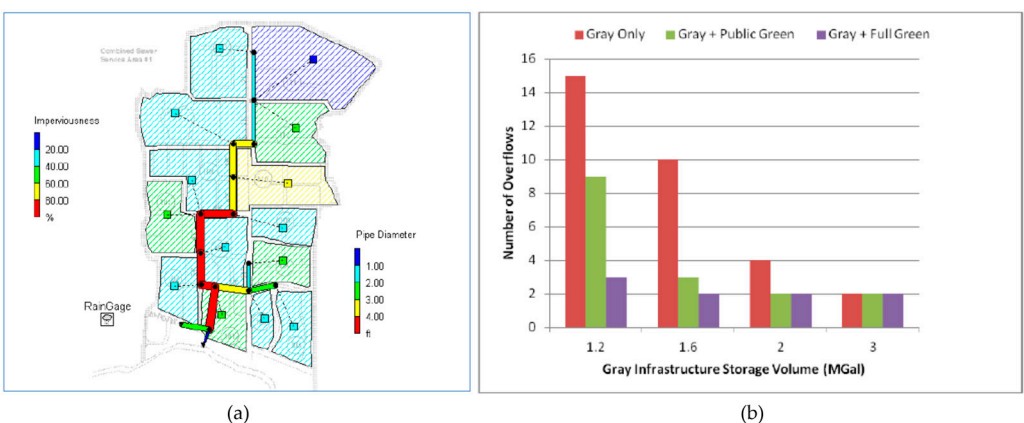

(a)  (b)

**Figure 2.** SWMM data of CSOs with Feasibility. (**a**) Hypothetical Sewershed Model (**b**) Gray and Green CSO Controls. Reprinted/adapted with permission from Ref. [18].

*3.2. Green Infrastructure*

According to the EPA, green methods, as an inexpensive solution, should be included in any CSO reduction plan. Stormwater drainage is achieved through pervious roads and parking lots, creating green spaces, rain gardens, and bioswales. Retention can be achieved with green roofs, retention ponds, flood plains, and planter trenches. Municipalities using this method often make a call to action in their communities to enact green methods around homes and businesses. They also typically incentivize businesses to use green site plans by creating award programs, lowering permitting fees, taxes, and stormwater fees, and offering grants [24]. Green infrastructure adds value to improving communities by adding green spaces and beautifying public areas, which is not applied with most gray infrastructure improvements. However, it requires matching public interest and government aspirations for a healthy environment, which is critical to adequate environmental protection and sustainability. When the two elements work together, they create a more potent force for positive change through green infrastructure.

These low costs and added benefits are why the city of Philadelphia focuses 70 percent of its LTCP efforts to curtail CSOs on a USD 1.7 billion green infrastructure initiative, including all these methods [25]. SWMM data suggests they have a limiting control of CSOs and that green infrastructure cannot be the only method used [17]. The goal of Green City, Clean Waters is to increase green stormwater infrastructure in Philadelphia to make it a significant portion of the EPA-mandated goal to reduce the amount of polluted stormwater overflows discharging into the creeks, streams, and rivers in and around the city by 85% by 2035 [26]. In most cities, green infrastructure is a secondary or tertiary part of the LTCP.

### 3.3. Gray Infrastructure

Gray infrastructure for stormwater management refers to a network of water retention and purification infrastructure (such as pipes, ditches, swales, culverts, and retention ponds) meant to slow the flow of stormwater during rain events to prevent flooding and reduce the amount of pollutants entering waterways [27]. Gray infrastructure improvements are often preferred because they provide a traditional and engineered system with reliability, proven technology, immediate impact, and regulatory compliance. The objective of gray infrastructure during CSOs is to create more storage in the system on the way to the WWTP or to increase the treatment plant's capacity. Due to the age of combined sewer systems, CSOs are exacerbated by cracked, damaged, and improperly fitted pipes. The inflow of water at poor connections or infiltration into pipes through cracks can substantially add to flow during a storm event. Another concern with old pipes is that the internal smoothness changes with scouring and scale buildup over time. These pipes can be replaced or lined with a smoother material that lessens the diameter but increases flow. Through the repairs and maintenance of deteriorated pipes, the capacity of the sewer system can be improved. Further capacity can be gained by increasing the sizes of pipes, especially the interceptors that run perpendicular to CSO outfalls. Adding capacity can also be achieved by adding retention basins or underground storage. One example is the highly ambitious USD 3.8 billion Tunnel and Reservoir Plan (TARP) project in Chicago. The TARP, also known as "Deep Tunnel," is a system of deep, large-diameter tunnels and vast reservoirs designed to reduce flooding and improve water quality in Chicago-area waterways.

It shields Lake Michigan from sewage overflow-related pollution [28]. Large underground tunnels have added 2.3 billion gallons of extra capacity to the system, with 110 miles of length completed in 2006. Since the tunnels became operational, CSOs have been reduced from an average of 100 days per year to 50 days. Three reservoirs with a 15.15-billion-gallon capacity will be completed by 2029. CSOs will be eliminated from their service area after the completion of TARP. Figure 3 shows how TARP works. The tunnel system captures the "first flush," the highest pollutant-loaded combined sewer flow. The reservoirs have gone a step further and collected more CSO volume. Many cities are adding similar elements on a smaller scale. SWMM data has shown that gray infrastructure can be a stand-alone method or work in conjunction with green infrastructure [17].

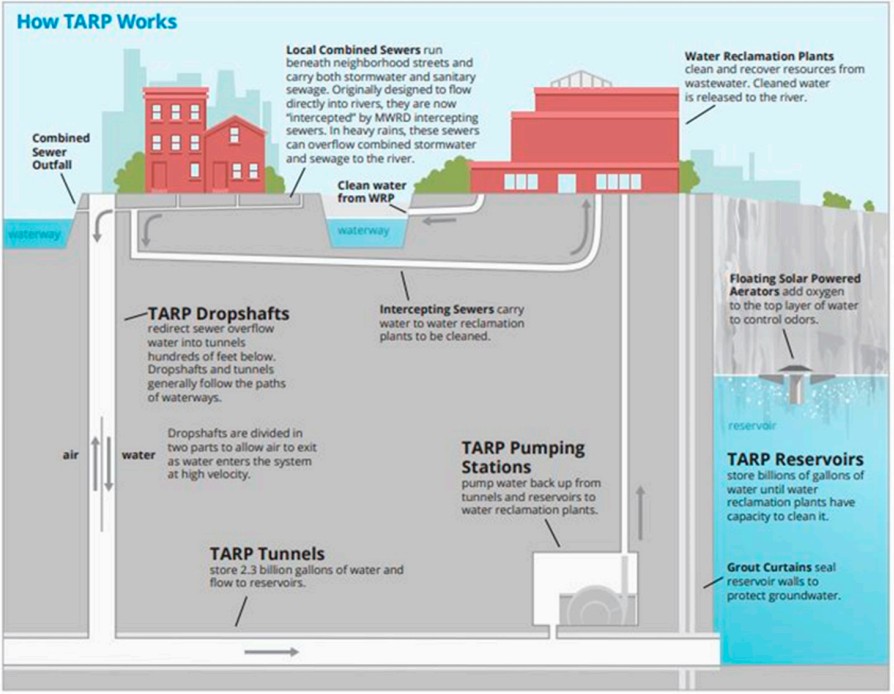

**Figure 3.** Illustration of TARP. Reprinted/adapted with permission from Ref. [25].

## 4. Smart Sewerage Control

These days, many utilities are applying advanced technology and data analysis to traditional sewage systems. In addition to the physical construction of the sewer system, technological integration plays a significant role in improving the efficiency, safety, and sustainability of wastewater and stormwater management. Typical smart technologies for utility include sensor networks, data analytics and artificial intelligence, water leak detection, flood detection, and remote monitoring and control. A smart sewage system with these features can become a component of a smart city by integrating with other city infrastructure and services.

Greece is considered the forerunner of modern sanitary systems. Modern sanitation systems have been characterized by the fast removal of waste and stormwater since the second half of the 19th century in European and American cities [29]. The 20th century marked a transformative era in the realms of wastewater management, environmental science, and societal attitudes concerning pollution. During this period, governments started enforcing wastewater treatment requirements. Before World War I, which led to the interruption of the installation of wastewater treatment facilities, they were constructed in the main cities of Europe [30]. One of the earliest modern-era WWTPs in Europe was the sedimentation treatment plant in Bubeneč (Prague), built in 1900–1906 [29]. Municipal wastewater treatment facilities (WWTPs) consistently oversee and gather data from unit processes and operations. Due to widespread interest in big data integration in WWTP, advancements in data-driven process control and performance analysis could allow the wastewater treatment industry to reduce costs and improve operations [31]. However, there is no single approach to developing smart sewers because utilities can integrate the technology that best benefits their system type, size, and performance goals [32]. Smart sewer technologies can be grouped into three main categories: (1) asset management tools, (2) system monitoring and operations tools, and (3) system condition and capacity tools [33].

In adverse weather conditions, an emerging control method may be able to control CSOs as situations change within the system. Wireless-enabled sensors, such as weirs, gates, pumps, and valves, are strategically placed in the wastewater system. The primary goal of using wireless-enabled sensors and smart control systems in wastewater management is to optimize the use of available capacity by strategically moving wastewater within the system. The system can effectively manage wastewater flow, prevent overloading in certain areas, and make best use of the existing infrastructure. During storm events, certain parts of the CSS can become overwhelmed due to the sudden increase in water flow, leading to CSOs where untreated or partially treated wastewater is discharged into the environment to prevent flooding. However, other areas within the system may have spare capacity and can handle additional flow without causing overflows. By leveraging wireless-enabled sensors, smart control can identify these areas with available capacity. This technology solves the problem by redistributing flow in pipes close to capacity and directing it to other pipes with free space, simplifying integrated sewage systems and enabling more sophisticated real-time monitoring capabilities.

The integration of intelligent control technologies, such as predictive modeling, real-time control, and artificial intelligence, synergistically collaborates to manage CSOs effectively. For instance, the predictive models utilized to estimate the volume of wastewater within the sewer system undergo regular updates to incorporate newly available data derived from precipitation forecasts [34]. At the same time, real-time control intervenes in the system to adjust flow direction toward existing storage or other water control facilities [34]. This sophisticated control is accomplished by specific algorithms adopted by individual governmental entities. Algorithm examples include a decentralized real-time control algorithm, a fuzzy-based real-time control algorithm, and a particle swarm optimization algorithm, to name a few [35–37]. Artificial intelligence (AI) can be implemented through two different approaches: model-based AI and data-driven AI, with the choice depending on the specific case. The concept of model-centric AI pertains to the iterative enhancement

of an artificial intelligence system by focusing on refining an established model without altering the quantity or structure of the gathered data. In contrast, proponents of data-centric AI adhere to a consistent model while continuously improving the data's quality [34].

Despite the presence of smart sewer technologies that enhance the management of CSOs, there are still notable challenges and constraints that need to be addressed. Due to the unfavorable conditions present in sewage structures, wireless technology necessitates hardware components that exhibit durability. Additionally, the battery utilized in the wireless device necessitates replacement, a task that mandates the sewerage personnel to gain entry to the challenging environmental conditions within the structures [38]. In this regard, the unfavorable conditions within sewage infrastructure also pose obstacles to transmitting real-time data, primarily attributed to the presence of chemical and biological impurities in the structure. [39]. One other significant technological obstacle is the existing disparity between the timeframe for optimizing intricate networks and the timeframe for implementing necessary actions when making network management decisions in practical settings [40]. As processing power continues to advance, the potential for decreased calculating time in solving optimization problems is conceivable. However, the shift towards more complex stormwater management systems that encompass multiple functions will significantly increase the computational cost. This is due to the need for concurrent modeling and trade-off analysis between various objectives [40].

## 5. Case Study of South Bend, Indiana

In South Bend, Indiana, there are 36 outfalls, and the city processes a daily volume of 48 to 77 million gallons of water [41]. The average yearly rainfall in this region is 38 inches [42]. From 1990 to 2004, South Bend experienced an average of 2 billion gallons of CSOs and invested USD 87 million in CSO infrastructure. In 2004, the city submitted its LTCP to the EPA to address this issue. The plan involved separating some CSO outfalls, upgrading the WWTP, and installing nine underground storage tanks across the city. In 2008, the City of South Bend embarked on a transformative initiative known as the CSOnet project. This project proved highly successful, prompting the city to update its long-term control plan to incorporate the CSOnet system. CSOnet is a cutting-edge real-time decision support system, granting the City of South Bend a profound understanding of its sewer system while providing greater control and optimization capabilities. With CSOnet in place, the city can now effectively minimize sewer overflows into the river and fully maximize the capacity of its existing infrastructure, a significant environmental achievement. In 2017, Phase I of South Bend's LTCP was completed, which involved several critical tasks. The plan included separating numerous sewers, strategically installing throttle valves at various outfalls, and integrating an impressive network of 150 sensors and 12 actuators. These sensors and actuators effectively control nine pumps and three weirs, adding sophisticated management and efficiency to the city's sewer system. Phase I costs exceeded USD 150 million, but the CSOnet improvements cost only USD 6 million.

CSOnet brings three significant benefits to the city's sewerage management system. Above all, the entire integrated sewage network can be monitored in real time. This continuous monitoring allows any potential problem or anomaly to be immediately detected and addressed. Second, CSOnet optimizes water flow at CSO outlets to ensure these outlets operate at peak efficiency, reducing the risk of sewage overflow and minimizing its environmental impact. Third, the system is proactive by strategically draining clean reservoirs prior to expected heavy rainfall events. This preemptive measure creates additional capacity within the system, mitigating the risk of overloading the sewerage infrastructure during heavy rains. To achieve this, it collects data from an extensive network of 150 sensors that forward the information to 17 gateways. Data is transferred to a central database using cloud-based technology. This database allows data to be collected and shared in real time. The architecture of CSOnet revolves around a local Wireless Sensor and Actuator Network (WSAN) interfacing with an existing Wide Area Network (WAN) via gateway devices, as shown in Figure 4. CSOnet, therefore, is viewed as a heterogeneous sensor-actuator network

consisting of four types of devices: (1) Instrumentation Node, (2) Relay Node, (3) Gateway Node, and (4) Actuator Node [43,44]. CSOnet's I Node and R Node are both based on the Chasqui processor module [44]. The distinction between these two devices is that the I Node incorporates a sensor management subsystem. In contrast, the R Node does not necessitate such a subsystem, as its sole function is to relay data. I Nodes are typically positioned within the manholes of the sewer system. An I Node is affixed to the manhole cover and linked via a cable to a pressure or flow sensor within the sewer conduit. Subsequently, the I Node transmits the sensor data from the sewer manhole to an R Node, typically on a traffic signal or utility pole. The A Node is responsible for regulating the flow within the interceptor line. At the same time, the G Node acts as a conduit connecting this specific WSAN to adjacent WSANs along the interceptor line, both upstream and downstream.

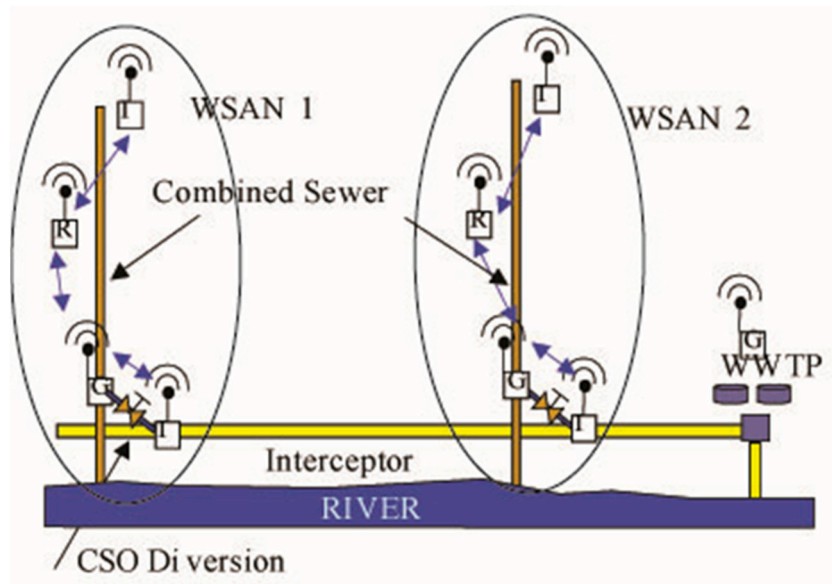

**Figure 4.** CSOnet's Architecture. Reprinted/adapted with permission from Ref. [44].

The city leverages IBM's Intelligent Operations Center software, along with supervisory control and data acquisition (SCADA) information and the city's geographic information system (GIS), to visualize and understand data. Integrating these technologies creates a comprehensive interface that gives public works departments a real-time view of the entire pipe network. Within this geospatial representation, public works teams can easily access important information such as current pipe levels, watershed levels, remaining storage capacity in basins, and CSO occurrences. Enhanced situational awareness allows cities to proactively manage their sewage systems and respond effectively to urgent problems or incidents.

Numerous improvements could come from these observations. Many dry-weather CSOs and backflow often originate from blockages in a pipe. These blockages can appear due to the intrusion of tree roots or the accumulation of debris. The financial impact of dry CSOs is significant, given their unpermitted status and associated substantial fines. The repercussions extend to property damage, incurring multimillion-dollar city costs annually. Overtime data can help provide a comprehensive understanding of expected flow in pipes under normal and peak conditions. If a pipe exceeds established normal levels, a city can effectively establish a proactive maintenance protocol to prevent overflow and backup by dispatching staff to rectify blockages. This preventive maintenance system is essential, instead of fixing pipes after overflows and backups. In particular, South Bend's implementation of this approach has had remarkable results. The system helped the utility eliminate critical dry CSOs, which occurred an average of 27 times per year, by isolating maintenance issues in the first year of operation [45].

Another critical benefit is inflow and infiltration (I/I) management. I/I can be identified through the events of higher-than-anticipated flows during storms or heavy rains. The final advantage encompasses the utilization of collected data for system operations, such as SWMM or analogous modeling tools. These applications facilitate a more profound understanding of the system dynamics and deficiencies in capacity. This data-driven insight empowers municipalities to judiciously allocate resources toward critical gray infrastructure investments [46].

In South Bend, various sensors form part of the efforts to prevent CSOs at outfalls along the St. Joseph River. These outfalls are outfitted with weirs that impede the entry of wastewater into the river. Instead, the wastewater is rerouted into a throttle line intricately linked to an interceptor leading to the wastewater treatment plant. In the case of moderate rainfall, an interesting observation emerged. Some outfalls remained free of overflow, while others experienced it. This discrepancy suggested that the interceptor had not yet reached its maximum capacity. The potential solution was to arrest the flow of stormwater at the main lines of non-overflowing outfalls. This approach would generate additional capacity within the interceptor, facilitating faster drainage for those trunk lines that were indeed overflowing.

Considering this notion, a series of valves were introduced to each throttle line, strategically positioned between the trunk line and the interceptor. The functioning of these valves was regulated by a competitive algorithm, granting them the ability to open and close in response to conditions. In the midst of a storm event, the throttle valves associated with trunk lines at full capacity are allowed to release their contents into the interceptor via controlled openings controlled by the CSOnet system. Meanwhile, the other throttle valves remain shut until each corresponding trunk line attains a predetermined parameter, signaling the valve to "compete" for access to the interceptor. This setup fosters a dynamic allocation of resources, optimizing the utilization of the interceptor and effectively managing CSOs during varying storm conditions.

Figure 5 shows how the algorithm would operate during a storm event. This algorithm also relies on real-time data but focuses on dynamically adjusting the flow rates and storage volumes of different parts of the sewer network to prevent overflows. CSOnet implements a control strategy aimed at regulating the flow of water directed into the interceptor sewer, originating from the combined sewer trunk line. Consequently, the control actuation takes place at the CSO diversion structure. Montestruque and Lemmon [43] simulated CSOnet's algorithm on the real South Bend interceptor sewer line, which consists of CSO diversion structures, with three different scenarios: (1) S1, 0.485 inches of rain in 11 h, (2) S2, 0.799 inches of rain in 13 h, and (3) S3, 2.046 inches of rain in 19 h. The simulation results show that the proposed controller reduces total storm overflows by 24−40% over existing fixed thresholding strategies. It uses predictive models to anticipate CSOs and make adjustments accordingly. The dynamic control strategy continuously gathers feedback from each time step [9].

The last significant benefit of CSOnet is that it is creating storage for stormwater before a storm event happens. A high stormwater flow event can be anticipated in accordance with weather prediction algorithms. It can trigger CSOnet to dynamically activate pumps to drain retention basins into the river before the storm occurs. Less stormwater gets into the sewer system by allowing more space for stormwater.

Since the implementation of CSOnet in 2008, along with the other Phase I CSO infrastructure improvements, CSOs have dropped from 2.1 billion gallons in 2008 to 458 million gallons in 2014. Phase 1 was completed in 2017. The smart sewer system provides data for various parameters, including flow, depth, velocity, and weir/gate control valve position. The system also contains smart moving valves that direct flow in the sewer and control stormwater basin levels. SWMM data suggests that CSOnet has removed over 75% of the annual CSO volume and prevented more than 1500 million gallons of combined sewage from entering the St. Joseph River annually [41].

Flow of Wastewater and Stormwater During the Middle to End of a Storm Event

**Figure 5.** The Competing Algorithm in Action.

## 6. Smart Swerage Control Application

Many cities still convey rainwater and wastewater through the same pipe network. A CSO occurs when excess rain overloads these pipes, and dirty water flows into the river. As federal regulation requires, many cities are upgrading their sewer systems to end this practice and improve local water quality to establish an LTCP. Rapid economic development, massive population growth, and global climate change generate tremendous pressure on infrastructure planning and development as a part of the LTCP. The LTCP encompasses the complicated strategic process of assessments and analysis, budgeting and financing, design and engineering, regulatory approvals, and construction and implementation for the functioning of a society or community. It also faces challenges such as funding limitations, bureaucratic hurdles, environmental concerns, and ensuring the equitable distribution of benefits. For instance, complying with federal regulations, such as the Clean Water Act in the United States, can be complex and time-consuming. Municipalities must navigate the intricacies of these regulations, which often involve stringent water quality standards and reporting requirements. In addition to federal regulations, municipalities must also comply with state and local permitting processes. These processes can involve multiple agencies and layers of approval, leading to delays and administrative burdens. Developing and implementing LTCPs often requires significant financial resources. Municipalities may struggle to secure the necessary funding through grants, loans, or rate increases, which can be met with resistance from residents and businesses. Additionally, budget limitations and limited internal capacity are the two most important constraints city administrations face in adopting smart technologies [47]. To surmount these obstacles, municipalities commonly enlist the aid of environmental consultants, legal experts, and regulatory bodies. They also proactively communicate with the public and other stakeholders to cultivate support and address concerns at the outset of the planning process. For all cities, local management innovations, including decisions regarding public engagement, will be critical in shaping future urban stormwater systems [48]. Therefore, fully maximizing the capacity of the existing infrastructure with smart control is considered a cost-effective and straightforward solution to reduce peak runoff and CSOs.

Richmond, VA, focuses on traditional methods using primarily gray and green infrastructure improvements, monitoring, and modeling. Research suggests that by combining current efforts with wireless sensor technology, Richmond can better meet its LTCP goals to prevent CSOs by inexpensively enhancing monitoring, dynamically controlling stormwater flow, and making more informed decisions on infrastructure improvements.

### 6.1. Application to the City of Richmond, Virginia

Richmond, Virginia, has faced tremendous challenges with combined overflows. According to the US Census Bureau 2020, Richmond has a population of over 226,000 and an area of 62.5 square miles [19]. Richmond receives an average of 43 inches of rain annually [49]. Parts of Richmond's first sewer system are over 150 years old and were designed as a CSS that services 18.8 square miles [50]. Richmond began addressing CSO problems back

in the 1970s. Richmond had 31 CSO outfalls along the James River or its tributaries, especially Shockoe Creek and Gillies Creek [51,52]. Richmond has implemented green infrastructure (i.e., rain gardens, green roofs, rain barrels and cisterns, and permeable pavement) and gray infrastructure. Gray infrastructure projects improve water quality, including replacing combined sewer pipes with separated pipes for wastewater and stormwater, building storage facilities to hold excess water, and increasing the capacity of the wastewater treatment plant. In its LTCP, Richmond's central strategy involved the construction of an extensive storage mechanism designed to capture the influx of stormwater blended with untreated sewage following rainfall [50,53]. According to the General Assembly Report [54], Richmond's CSS area covers 19 square miles and includes 25 combined sewer outfalls, as shown in Figure 6. Approximately 32% of the city's land area is within the combined sewer area, along with 52% of the city's population. In Figure 6, the seven black dots indicate closed outfalls from 31 active outfalls with investments in gray and green infrastructure and the RVA Clean Water Plan for monitoring and modeling.

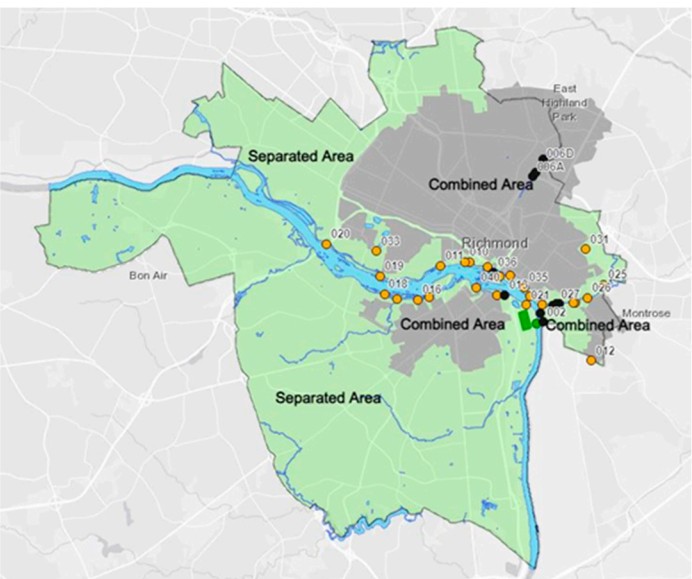

**Figure 6.** Richmond's Sewer System with Closed Outfalls in Black and Active Outfalls in Yellow Dots. Reprinted/adapted with permission from Ref. [50].

CSS improvements had occurred from 1970 to 2022; the Shockoe Retention Basin was completed in 1983 with a 50-million-gallon connection to the main combined sewer line. The basin can divert water with weirs just before the WWTP, contributing to minimal overflow at 10 CSO outfalls. Another major storage improvement implementation was adding an underground storage tunnel deep within an underlying granite layer. The Hampton-McCloy Tunnel, completed in 2003, stores 7 million gallons and has helped significantly lower CSOs in three outfalls near an important recreational area. After completing Phase 2 from 1988 to 2002, Richmond could capture 85% of CSO. The WWTP has been increased to treat 75 million dry flow gallons or 140 million gallons of storm flow per day with Phase 3 [55].

Richmond has placed importance on monitoring and modeling the sewer and stormwater systems. These methods include mapping the CSO areas, reviewing documents to show pipe lengths, material, and diameter, taking water samples, and monitoring CSO flow. The city's efforts have improved the CSO capture rate to approximately 91%. This means that CSO flowing into the James River has been reduced by more than 3 billion gallons a year [54]. Richmond mapped out a three-phase plan in its LTCP to the EPA. Currently, phases I and II are complete, including the Shockoe Basin and the Hampton-McCloy tunnel. In Phase III, many implementations are planned, such as increasing storage capacity, upgrading the wastewater treatment plant, installing green infrastructure, educating the

public, and planning for the future. The CSO infrastructure improvement plan includes increasing the wet weather treatment capacity to 300 million gallons per day, disconnecting two outfalls, increasing the interceptor line in the lower Gillies Area, replacing a regulator and adding a million gallons of storage at a problematic outfall, adding 15 million gallons of storage to the Shockoe basin, and chlorine disinfection at another problematic outfall. Phase III green infrastructure plans include creating 18 acres of impervious surfaces. It can be completed by improving 6 acres of public utility property, 2 acres of city-owned vacant properties, and 2 acres of public parks, and installing 24 tree boxes of 8 acres total drained to the Combined Sewer System area [56]. Data collected from SWMM shows a ten-million-gallon annual drop in CSOs from all the green infrastructure implementations and a drop from 1.67 billion gallons to 228 million gallons from the gray infrastructure improvements.

### 6.2. Cost Comparison with SWMM

EPA's SWMM is widely used for the planning, analysis, and design related to stormwater runoff, combined and sanitary sewers, and other drainage systems, mainly for the frequency of CSO into receiving waters [57]. However, this modeling has limitations in modeling large-scale, non-urban watersheds, with no application to a forested area or irrigated cropland [57,58]. Achieving model realism, which involves ensuring that the model reasonably and accurately reflects real-world conditions, is associated with two distinct elements within the model: initial conditions and the structural equations of the model [59]. It can be used to evaluate gray infrastructure stormwater control strategies, such as pipes and storm drains, and create cost-effective green/gray hybrid stormwater controls [17]. SWMM data can identify the number of gallons of CSOs that various methods can prevent. By dividing these figures by projected expenses, it becomes possible to determine the cost incurred per million gallons of yearly CSOs. Table 1 shows a substantially lower cost for the city of South Bend's wireless sensor method than Richmond's gray and green methods or Grand Rapid's total separation.

**Table 1.** Cost Comparison with Different Methods to Manage CSOs.

| City | CSO Control Method | Cost | Drop in CSOs | Cost Per Million-Gallon Annual Drop in CSOs |
|---|---|---|---|---|
| Richmond, VA | Phase III—Gray Infrastructure | USD 375 M | 1.44 billion gallons | USD 260,000 |
| Richmond, VA | Phase III—Green Infrastructure | USD 2.6 M | 10 million gallons | USD 260,000 |
| Grand Rapids, MI | Sewer Improvement Project—Total Separation | USD 400 M | 629 million gallons | USD 636,000 |
| South Bend, IN | CSOnet—Wireless Sensor Technology and Throttle Line | USD 6 M | 312 million gallons | USD 19,200 |

### 6.3. The Benefit of Integrating Wireless Technology in Richmond, VA

Richmond has implemented diverse strategies to mitigate CSOs, aiming to manage the intricate interplay between human activities, rainfalls, snow melts, inflow and infiltration, pipe obstructions, floods, river currents, and evolving weather patterns. Despite investing significant amounts, exceeding hundreds of millions of dollars, into initiatives like green infrastructure, sewer upkeep, facility upgrades, and expansive subterranean reservoirs, the city finds a static solution for CSOs. It is considered a passive system in which all components are set in place to control dynamic forces. While the system works well on most days, it still does not entirely stop the millions of gallons of CSOs that occur on many days. Richmond's CSOs only happened for 94 days in 2022, adding up to 262 billion gallons [55]. Designers of the static system have to overcompensate to allow the system to handle the higher peaks of the SWMM hydrographs.

An example is shown in a cost-versus-benefit study to determine the value of increasing the interceptor size in Richmond's Gillies Creek area. It was determined through SWMM that it would cost USD 300 million for storage that would only be at full capacity once every five years based on 5-year storm models. Preparedness for a five-year storm is not even in the scope of an EPA control policy. At this gray infrastructure level, costs

increase exponentially for small percentage gains. In static systems, achieving 100 percent control is almost impossible [46]. Richmond's LTCP only plans for a 92% reduction in CSOs [55].

Integrating wireless sensors into Richmond's LTCP could bring substantial advantages. The city's efforts to manage CSOs have hinged on activities like monitoring, data collection, and modeling. Enhancing the ability to manage CSOs is possible through the utilization of real-time monitoring, effectively harnessing Richmond's existing SCADA and GIS capabilities in conjunction with the technology. By incorporating monitoring, the reduction of Infiltration and Inflow (I/I), obstructions, and dry weather CSOs can be accomplished. Furthermore, optimizing the system to ensure that only stormwater enters through drains could augment its capacity significantly.

A pivotal strategy involves optimizing the city's sewer infrastructure. Leveraging data collection, Richmond can gain invaluable insights into identifying pipes requiring lining or replacement. Richmond has many CSO locations that can be improved by controlling flow and creating more capacity. With the strategic integration of wireless sensors and pumps, the flow redirection from problematic CSO outlets to the nearby dual 7-foot diameter interceptors is feasible. Notably, the majority of CSOs occur at the outfalls that drain to smaller interceptors. These areas can improve incredibly effectively by adding actuated valves and an added throttle line similar to South Bend.

Using SWMM of CSO data gathered between July 2017 and June 2018, as documented in Richmond's CSO monthly reports, a striking resemblance emerges between Richmond and South Bend's predicament. CSOs are happening at some trunk line outfalls that share an interceptor but not at others. It can be observed in Figure 7.

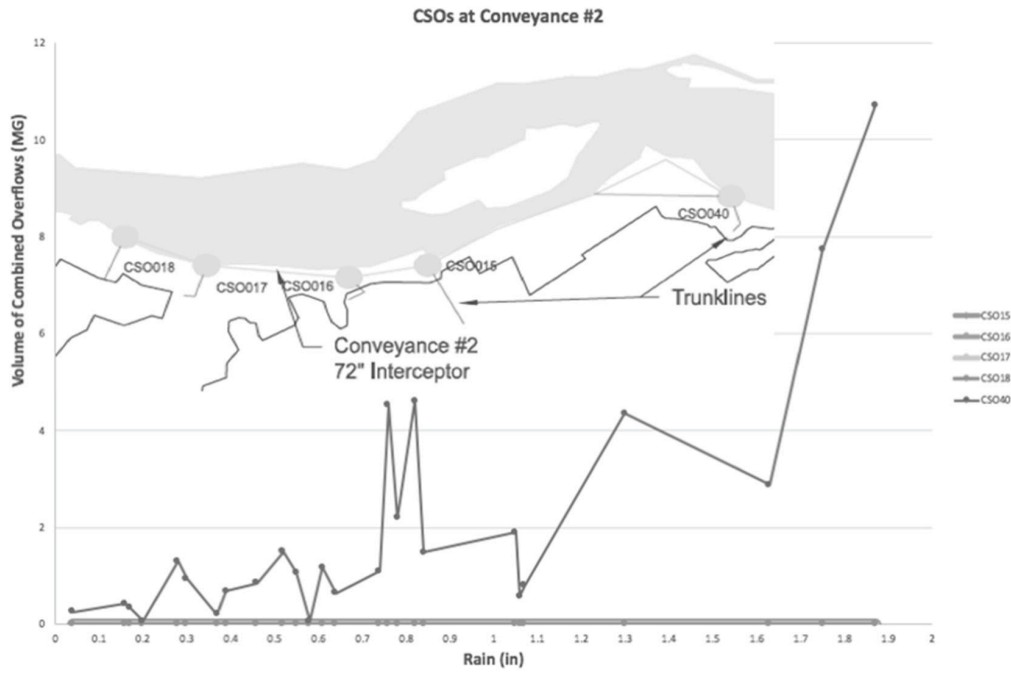

**Figure 7.** Overflow Activity at an Interceptor called Conveyance Number 2.

Both cities experience CSOs at specific trunk line outfalls that share an interceptor, while other points in the network remain unaffected. The correlation underscores the potential for Richmond to adapt successful strategies from South Bend to ameliorate its CSO challenges. Only the outfall "CSO40" at the end of the interceptor has CSOs during rainfalls of less than 2 inches. Controlling flow in the Gillies Creek area can also help Richmond lower the cost of the USD 300 interceptor as part of Phase III of the LTCP.

It should be reminded that, as described earlier, potential technological challenges and limitations on wireless technology exist. It is true that the evident advantages of these

advancements in technology can also pose substantial obstacles for water agencies and service providers. The large-scale implementation of such technologies necessitates a high level of assurance in the sustained functionality, upkeep, and compatibility of the installed systems [40]. Therefore, it is imperative to establish standards or protocols and ensure their widespread adoption, and this will instill the necessary long-term confidence in the sector and facilitate more predictable market access for future technical advancements [40].

## 7. Discussion and Conclusions

Numerous municipalities are actively addressing the EPA's requirements for controlling CSOs, employing diverse strategies to bring CSO levels within acceptable limits. While separation has demonstrated the most successful CSO control, financial constraints render this option unfeasible for many urban centers. As a result, most cities are compelled to manage CSOs through augmentations or adaptations to their existing infrastructure. Predominantly, these efforts have leaned heavily on the SWMM, employing a foundational stormwater flow as a baseline for predicting CSO changes. It informs the city on how and where implementations should be made. Nonetheless, this static system yields a somewhat rigid framework designed to support combined stormwater and wastewater flow during high peak flow events. This methodology guides municipalities in pinpointing strategic implementation areas to reinforce the system's capacity for handling peak flow scenarios. Integrating gray and green infrastructure has yielded modest successes, albeit at a considerable cost escalation, as the pursuit of more significant percentage gains continues.

In this circumstance, wireless technology emerges as a dynamic solution to address the intricacies of these intricate and turbulent systems. The capacity for real-time monitoring introduces a level of seamlessness in data collection, fostering the creation of more precise models if it is operated as intended. The consequential benefit is enhanced decision making for future projects. Real-time modeling has exhibited remarkable efficacy in curbing unauthorized dry CSOs and promptly identifying system deficiencies. Wireless sensor technology has effectively averted CSOs by leveraging existing infrastructure at a fraction of the cost of alternative methods.

South Bend's case serves as a testament to the cost-effective potential of this technology in CSO control. Nevertheless, it becomes evident that wireless sensor technology alone is not a comprehensive solution. The optimal CSO reduction is achieved by its integration with other control methods. Similarly, Richmond, a comparable municipality, can benefit substantially from this technology. Abundant untapped capacity resides within their current infrastructure, fortified by a robust commitment to ongoing data collection. The real-time monitoring potential can significantly enhance modeling accuracy, leading to well-informed decisions regarding gray and green infrastructure enhancements. Not only Richmond, but many cities struggling with CSO problems will have significant cost savings compared to other CSO control methods. Integrating existing infrastructure problems by adopting new technologies can benefit current issues and LTCPs by lessening financial constraints, regulatory requirements, and bureaucratic hurdles.

Richmond's earnest endeavors in CSO management, approaching the halfway mark of their goal, are poised to evolve further as they embark on Phase III of their LTCP. The analysis of SWMM data signals the potential for more effective CSO control through the assimilation of wireless sensor technology. This approach, in essence, could serve as a transformative leap in their ongoing CSO mitigation journey. The EPA's SWMM is used for planning.

In summary, the establishment of collaborations among significant players to formulate initial objectives and address areas of interest becomes a more crucial aspect within the CSS framework, as the operation of sewer systems and the reduction of CSOs constitute an incredibly intricate domain, necessitating the adoption of an integrated strategy tailored to each unique scenario. In this regard, future research should focus on a new system that holistically entails the process of problem analysis, the establishment of objectives, and proposes possible solutions. This would necessitate a substantial understanding of

computer science, electrical and civil engineering, and other related areas because this comprehensive knowledge is essential for the development of integrated, multifaceted solutions and advancements in technology. Moreover, an interdisciplinary methodology is essential for the smart construction, management, and enhancement of climate-resilient and durable wastewater infrastructure, which is capable of safeguarding the well-being of both humans and ecosystems even in the most demanding circumstances.

**Author Contributions:** All authors contributed to conceptualization of the manuscript; methodology, Y.E.J., M.M.J. and T.S.; software, Y.E.J. and T.S. validation, Y.J, M.M.J. and H.J.; formal analysis, Y.E.J. and M.M.J.; investigation, Y.E.J.; data curation, Y.E.J. and M.M.J.; writing-original draft preparation, Y.E.J.; writing-review and editing, Y.E.J., M.M.J. and H.J.; visualization, Y.E.J.; supervision, M.M.J. and H.J.; project administration, Y.E.J.; funding acquisition, Y.E.J. and H.J. All authors have read and agreed to the published version of the manuscript.

**Funding:** The APC was funded partially by the Seoul National University of Science and Technology.

**Data Availability Statement:** Data are available upon request.

**Acknowledgments:** This work was supported by the National Research Foundation of Korea (NRF) grant funded by the Korea government (MSIT) (No. RS-2023-00259995).

**Conflicts of Interest:** The authors declare no conflict of interest.

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
