# Peer review of "Contemplation of Improvement Efforts to Manage Combined Sewer Overflows†"

_infrastructures, doi:10.3390/infrastructures8100150_

Round 1

Reviewer 1 Report

This research work is on a topic of relevance and general interest to the readers of the journal. In the light of the above, there are no specific grounds for opposing the publication of this research work however few minor revision are needed to improve the impact of its content. As a first example, the title is okay but I think that it could be more effective.  In general, a proper discussion of the state of art is missing in the introduction. In particular, it could be a great opportunity to add a few example specifically related to the discussed topic carried out in other countries. I suggest giving a look at the latest published results of research studies about SuDS implementation in urban catchments: https://doi.org/10.1016/j.scs.2023.104856; https://doi.org/10.1016/j.uclim.2023.101596.  They are just an example but a wide variety of results had been obtained so far. The presentation and length is satisfactory, the overall readability is quite excellent, as well as the English translation, so I do not suggest any modification, reductions or deletions. I appreciated also the non-conventional structure of the paper, respectful of the content itself. I think that the discussion section would benefit from a deeper discussion and comparison with literature examples. 

Minor editing of English language required

Author Response

Thank you for your kind guidance and introduction to the latest literature. In the introduction section, we include recent study efforts, such as Stormwater Detention Tanks and the utilization of the Precipitation Variability Adaptation Index (PVAI) in SuDS. 

Reviewer 2 Report

The paper presents an in-depth exploration of smart sewerage control, particularly its application in managing CSOs in urban environments. The authors have highlighted the significance of integrating technology and data analysis to improve traditional sewage systems. The case studies of South Bend, Indiana, and Richmond, Virginia, provide valuable insights into the practical implementation of smart sewerage control systems. The discussion on the benefits of wireless sensor technology is noteworthy, and the cost comparison with the SWMM model adds depth to the analysis. Overall, the paper contributes to the understanding of how smart technologies can enhance wastewater management.

Specific Comments:

Introduction:

The introduction effectively lays out the importance of modernizing sewage systems using advanced technologies. To further engage the readers and create a stronger foundation for your argument, consider providing a succinct overview of the limitations or challenges posed by traditional sewer systems. By highlighting the shortcomings of existing methods, you can create a clearer context for why advancements and innovative approaches are essential. This additional insight will not only emphasize the need for technology integration but also enhance the readers' understanding of the issues your research addresses. The integration of smart sewerage systems into smart cities is a relevant point that sets the tone for the rest of the paper. Building upon this, providing a glimpse of how these technologies align with the broader goals of creating sustainable, efficient urban environments can add more depth to the introductory section.

Smart Sewerage Control:

The section provides a comprehensive overview of smart technologies applied to sewerage systems. To further enrich this section, you might consider including a concise overview of the technological landscape that led to the development of smart sewerage control systems. This could include a brief historical perspective on the evolution of wastewater management technologies and how the integration of data-driven approaches has revolutionized the field.

Clear explanations of various components such as sensors, repeaters, gateways, and actuators help readers understand the technical aspects. To enhance reader engagement, you could provide illustrative examples or analogies that simplify the understanding of these technical components. Additionally, consider including any recent advancements or emerging trends in these components that could potentially shape the future of smart sewerage control.

The description of the technology's ability to identify areas with available capacity and optimize wastewater flow is well-presented. To provide a more holistic view, you could delve into the methodologies or algorithms employed in this process. This could involve explaining how real-time data analysis and predictive modeling contribute to the identification of areas with capacity and the optimization of wastewater flow. This level of detail can provide readers with a deeper understanding of the technical underpinnings of these capabilities.

While the benefits of using wireless sensors are highlighted, it's also valuable to acknowledge potential challenges or limitations associated with their use. Consider dedicating a brief subsection to discussing maintenance requirements, potential interferences, or data security concerns. Addressing these aspects will provide a well-rounded perspective and demonstrate a comprehensive understanding of the technology's implications.

Case Study of South Bend, Indiana:

The case study of South Bend offers valuable insights into the practical implementation of a smart sewerage control system.

Clear presentation of data regarding CSOs, rainfall, and infrastructure improvements helps support the success of the CSOnet system.

The description of CSOnet's benefits is well-structured and highlights the real-time monitoring capabilities and optimization of water flow.

Including specific examples of how CSOnet's algorithms respond to storm events adds depth to the understanding of its operation.

Smart Sewerage Control Application:

The discussion on the challenges faced by cities in upgrading sewer systems and implementing LTCPs is informative. To provide a more comprehensive view of these challenges, you could briefly touch upon the regulatory and bureaucratic hurdles that municipalities often encounter. Additionally, mentioning any financial constraints that cities might face when adopting new technologies can emphasize the need for cost-effective solutions like smart sewerage control.

The application to the City of Richmond, Virginia, is well-detailed and highlights how wireless sensor technology can be integrated into existing strategies. To enhance the practicality of this application, you might consider providing specific examples of how the wireless sensors were strategically placed within Richmond's sewer network. This could include discussing the decision-making process behind sensor placement, considering factors such as high-risk CSO areas or locations prone to I/I.

The comparison with SWMM data effectively demonstrates the potential cost-effectiveness of wireless technology. To strengthen this comparison, consider addressing the limitations of the SWMM model itself. While it's a widely used tool, mentioning its assumptions and any discrepancies between its predictions and real-world outcomes can provide a more nuanced perspective.

Exploring the potential challenges or drawbacks of integrating wireless technology, such as privacy concerns or technical issues, could enhance the analysis. In addition to privacy and technical aspects, you could also discuss potential community acceptance of the technology. Exploring the human and social dimensions of adopting smart sewerage control can provide a holistic understanding of its implementation.

Discussion and Conclusion

Consider expanding on the discussion of potential future research directions in the field of smart sewerage control. While the paper excellently showcases the practical implementation of wireless sensor technology in managing CSOs, delving into areas that warrant further exploration can stimulate academic curiosity. For instance, you could discuss the feasibility of integrating artificial intelligence or machine learning into these systems to enhance predictive capabilities or explore ways to optimize sensor deployment for maximum coverage and efficiency.

Furthermore, discussing the potential transferability of the CSOnet system and similar technologies to other geographical locations or city sizes could open up avenues for international collaboration and broader application. This section could also touch upon the potential for interdisciplinary collaborations between urban planners, engineers, data scientists, and social scientists to create a more holistic approach to smart sewerage management.

I want to emphasize that my intention is solely to provide constructive feedback aimed at improving the quality of your work. Many of the suggestions I've provided may be open to debate, but I encourage you to carefully consider them as you continue refining your research. Please feel free to focus on the overall insights and ideas rather than providing detailed point-by-point responses to my comments. Your careful consideration of these suggestions can contribute to the enhancement of your research. Best of luck in further developing your work!

The overall use of English in the paper is clear and effectively conveys the technical concepts and information. However, there are a few instances where sentence structures could be refined for greater clarity and flow. Reviewing the text for minor grammatical errors and ensuring consistent use of terminology would further enhance the readability and coherence of the paper. Overall, the language is understandable and proficient, contributing to the effective communication of the research findings.

Author Response

Thank you very much for your comments. 

We revised the paper based on your comments as follows: 

Introduction:

  • The limitation of the CSS is essentially singular: polluted water going into the waterbody. We revised the introduction by adding the following sentence to re-emphasize the need for new technologies. “In essence, a significant drawback of the conventional system is in its inability to effectively segregate contaminated wastewater from the combined water during periods of intense precipitation.”

Smart Swerage Control: 

  • A brief historical perspective of wastewater management technologies is included.

  • Included technical aspects of smart sewerage control in the section of Case Study of South Bend, Indiana
  • The technologies are further explained as follows: “The integration of intelligent control technologies, such as predictive modeling, real-time control, and artificial intelligence, synergistically collaborate to effectively manage CSOs. For instance, the predictive models utilized to estimate the volume of wastewater within the sewer system undergo regular updates to incorporate newly available data derived from precipitation forecasts. At the same time, real-time control intervenes on the system to adjust flow direction toward an existing storage or other water control facilities. This sophisticated control is accomplished by specific algorithms adopted by individual governmental entities. Algorithm examples include a decentralized real-time control algorithm, a fuzzy-based real-time control algorithm, a particle swarm optimization algorithm, to name a few. Artificial intelligence (AI) can be implemented in two different approaches: model-based AI and data-driven AI, with the choice depending on the specific case. The concept of model-centric AI pertains to the iterative enhancement of an artificial intelligence system by focusing on refining an established model, without altering the quantity or structure of the gathered data. In contrast, proponents of data-centric AI adhere to a consistent model while continuously improving the data's quality.”

Discussion and Conclusion

  • The following paragraph is added to further discuss potential future research and interdisciplinary collaborations: “In summary, the establishment of collaborations among major players to formulate initial objectives and address areas of interest becomes a more crucial aspect within the CSS framework, as the operation of sewer systems and the reduction of CSOs constitute an incredibly intricate domain, necessitating the adoption of an integrated strategy tailored to each unique scenario. In this regard, future research should focus on a new system that holistically entails the process of problem analysis, establishment of objectives, and proposing possible solutions. This would necessitate a substantial comprehension of computer science, electrical and civil engineering, and other related areas because this comprehensive knowledge is essential for the development of integrated, multifaceted solutions and advancements of technology. Moreover, an interdisciplinary methodology is essential for the smart construction, management, and enhancement of climate-resilient and durable wastewater infrastructure, which is capable of safeguarding the well-being of both humans and ecosystems even in the most demanding circumstances." 

Case Study of South Bend, Indiana

  • Added the architecture of CSOnet and simulation case to identify and understand a mechanism of smart system. 

Smart Sewerage Control Application

  • Add detailed descriptions for regulatory and bureaucratic hurdles that municipalities often encounter, as well as financial constraints that cities might face.

  • Add a simulation case on how to utilize for CSO cases. 
  • Include limitations and some discrepancies about SWMM.

  • The following paragraph is inserted to remind readers of the potential challenges: “It should be reminded that, as described earlier, potential technological challenges and limitations on wireless technology exist. It is true that the evident advantages of these advancements in technology can also pose substantial obstacles for water agencies and service providers. The large-scale implementation of such technologies necessitates a high level of assurance in the sustained functionality, upkeep, and compatibility of the installed systems. Therefore, it is imperative to establish standards or protocols and ensure their widespread adoption and this will instill the necessary long-term confidence in the sector and facilitate more predictable market access for future technical advancements."

Reviewer 3 Report

Dear Authors, work is good, such changes needed:

1) specify aim and tasks

2) add citations of relevant literature, e.g., from applied works such as Tamm et al, 2023 The intensification of short-duration rainfall extremes due to climate change – Need for a frequent update of intensity–duration–frequency curves AND Grinberga et al 2021 Treatment of Storm Water from Agricultural Catchment in Pilot Scale Constructed Wetland AND Tamm et al Contributions of DOC from surface and groundflow into Lake Võrtsjärv

3) Formatting of Fonts in Tables and Figures needed

Author Response

Thank you for your kind comments. 

  • We revised the Objective and Scope of Study section to specify and clarify aims and tasks. 
  • We referenced and quoted  "There will likely be a higher frequency and intensity of extreme rainfall in the future, leading to an increased risk of urban flash floods sentence" from "Tamm et al., 2023 The intensification of short-duration rainfall extremes due to climate change – Need for a frequent update of intensity–duration–frequency curves."  
  • We formatted tables and figures correctly based on the author's guide.

Reviewer 4 Report

The manuscript utilizes SWMM software and combines case studies to analyze the management of waterway overflow under the combined flow system, which has certain practical significance. However, there are several issues that need to be addressed.

1. The words in Figure 1 are too small and blurry.

2. The introduction lacks relevant content of improvement efforts to manage combined sewer overflows.

3. If possible, please add a schematic diagram of the location of the study area and briefly explain why this study area was chosen as a research case? What is the representativeness?

4. Please draw a simple schematic diagram for the "4. Smart Sewerage Control" section.

5. There is an error in the numbering of chapter titles in the text, such as "4. Smart Sewerage Control ", "4. Case Study of South Bend", Indiana", 4. Smart Swerage Control Application", etc. Please verify and modify.

6. Can quantitative analysis be added to the abstract and results sections to increase the scientific and readability of the article, as there are more specific values of economic and sewage treatment capacity in the article.

Minor editing of English language required.

Author Response

Thank you very much for your kind comments. We revised the paper based on your comments. 

  • Figure 1 was corrected to recognize its figures and words.
  • We added citations of relevant literature, such as Stormwater Detention Tanks and the utilization of the Precipitation Variability Adaptation Index (PVAI) in SuDS in the introduction,  Tamm et al, 2023 The intensification of short-duration rainfall extremes due to climate change – Need for a frequent update of intensity–duration–frequency curves, etc.
  • We revised the Objective and Scope of the Study to improve the justification for this study.
  • The numbers of the Section were corrected.
  • The abstract and conclusion parts include an economic perspective for using a smart sewerage system.

Round 2

Reviewer 3 Report

ready to launch